

# Research on image classification method based on improved multi-scale relational network

Wenfeng Zheng[1]    Xiangjun Liu[1]    Lirong Yin[2]

[1] School of Automation, University of Electronic Science and Technology of China, Chengdu, China
[2] Department of Geography and Anthropology, Louisiana State University and Agricultural and Mechanical College, Baton Rouge, LA, United States of America

## ABSTRACT

Small sample learning aims to learn information about object categories from a single or a few training samples. This learning style is crucial for deep learning methods based on large amounts of data. The deep learning method can solve small sample learning through the idea of meta-learning "how to learn by using previous experience." Therefore, this paper takes image classification as the research object to study how meta-learning quickly learns from a small number of sample images. The main contents are as follows: After considering the distribution difference of data sets on the generalization performance of measurement learning and the advantages of optimizing the initial characterization method, this paper adds the model-independent meta-learning algorithm and designs a multi-scale meta-relational network. First, the idea of META-SGD is adopted, and the inner learning rate is taken as the learning vector and model parameter to learn together. Secondly, in the meta-training process, the model-independent meta-learning algorithm is used to find the optimal parameters of the model. The inner gradient iteration is canceled in the process of meta-validation and meta-test. The experimental results show that the multi-scale meta-relational network makes the learned measurement have stronger generalization ability, which further improves the classification accuracy on the benchmark set and avoids the need for fine-tuning of the model-independent meta-learning algorithm.

## INTRODUCTION

Deep learning has made significant progress in computer vision fields, but only on the premise that they have a large amount of annotated data (*Ni et al., 2019*; *Vinyals et al., 2016*; *Zheng, Liu & Yin, 2021*). However, it is impractical to acquire large amounts of data in real life. As far as deep learning is concerned, fitting into a more complex model requires more data to have good generalization ability. Once there is a lack of data, deep learning technology can make the in-sample training effect good, but the generalization performance of new samples is poor. Inspired by the human ability to learn quickly from a small sample, many researchers have become increasingly aware of the need to study machine learning from a small sample. In recent years, small sample learning has become

Corresponding author
Lirong Yin, lyin5@lsu.edu

a very important frontier research direction in deep learning. Human beings can learn new concepts from a single or a few samples and obtain extremely rich representations from sparse data. This ability is attributed to people's ability to realize and control their learning process, called meta-learning (*Biggs, 1985*; *Ding et al., 2019*; *Ma et al., 2021*; *Yin et al., 2019*).

Up to now, research on learning with fewer samples can be divided into two aspects: 1. Generation model based on probabilistic reasoning; 2. Discriminant model based on meta-learning.

The pioneering work of low-sample learning can be traced back to the work of *Fe-Fei, (2003)* which defined the concept of low-sample learning: learning the new category by using one or a few samples of the image of the new category (*Fe-Fei, 2003*). Li Fei-Fei proposed a Bayesian learning framework in 2004. Andrew L. Maas et al. used the Bayesian network to capture the relationship between attributes in 2009, which can deal with near-deterministic relationships and include soft probabilistic relationships (*Kemp & Maas, 2009*; *Li et al., 2015*). *Rezende et al. (2016)* implemented the hierarchical Bayesian programming learning framework in the deep generation model based on feedback and attention principles. Compared with the shallow hierarchical Bayesian programming learning framework (*Lake, Salakhutdinov & Tenenbaum, 2015*; *Tang et al., 2020b*), this method has a wide range of applications. However, more data is needed to avoid over-fitting (*Rezende et al., 2016*). In 2017, Vicarious researchers came up with a probabilistic model of vision, based on how the visual cortex works, reporting up from level to level, and named it the recursive cortical network. With the same accuracy, the recursive cortical network uses only one-millionth of the deep learning method's training sample. It can be used to crack various variants of text captchAs after one training (*George et al., 2017*; *Li et al., 2020*).

Meta-learning, also known as learning to learn, refers to using previous experience to learn a new task quickly, rather than thinking about the new task in isolation. *Lake et al. (2017)* emphasized its importance as the cornerstone of artifical intelligence. Up to now, the research directions of using meta-learning to deal with small sample learning include: 1. Memory enhancement; 2. Measure learning; 3. Learn the optimizer; 4. Optimize initial characterization.

## Memory enhancement

Memory enhancement refers primarily to the use of cyclic neural networks with memory or temporal convolution to iterate over examples of a given problem, accumulating information that is used to solve the problem in its hidden activation layer or external memory. Considering that neural network Turing machines can carry out short-term memory through external storage and carry out long-term memory through slow weight updating (*Graves et al., 2016*; *Tang et al., 2021*), *Santoro et al. (2016)* proposed a memory enhancement network based on long and short memory network (LSTM) in 2016 (*Santoro et al., 2016*; *Tang et al., 2020a*). Next, *Munkhdalai & Yu, (2017)* proposed a meta-network, which is composed of a meta-learning device and a learning device, and at the same time, a memory unit is added externally. By using the gradient as the meta-information, the fast

weight and slow weight are generated on two time scales, and then the fast weight and slow weight are combined by the layer enhancement method.

## Measure learning

Metric Learning refers to learning a similarity measure from data, and then using this measure to compare and match samples of new unknown categories. In 2015, Gregory Koch et al. proposed the twin network (*Chen et al., 2020*; *Koch, Zemel & Salakhutdinov, 2015*). The recognition accuracy of the model on Onniglot data is close to human. *Vinyals et al. (2016)* proposed an end-to-end and directly optimized matching network based on memory and attention (*Li et al., 2017*; *Vinyals et al., 2016*). This network can quickly learn with few samples, and for new classes never seen in the training process, the trained model is not changed, and the test samples can be classified with a few calibration samples of each class. *Sung et al. (2018)* proposed the relationship network in 2018. Learning to get similarity measure is more flexible and can capture the similarity between features better than the artificially selected measure method, and good results are obtained on several benchmark data sets learned with few samples.

## Learn the optimizer

The learning optimizer refers to learning how to update the learner parameters, that is, learning the updating function or updating rules of the new model parameters. *Ravi & Larochelle (2016)* used LSTM as a meta-learner, and took the learner's initial recognition parameters, learning rate and loss gradient as LSTM states to learn the learner's initialization parameters and parameter update rules. *Yang et al. (2020)* introduced metric Learning based on the method proposed by Ravi and Larochelle, and proposed a meta-metric learner. The author integrates the matching network and the method of updating rules using LSTM learning parameters, and obtains a better method. However, such a structure is more complex than the structure based on measurement learning alone, and each parameter of the learner is updated independently in each step, which will largely limit its potential (*Sung et al., 2018*).

## Optimize initial characterization

Optimize the initial representation, that is, optimize the initial representation directly. *Finn, Abbeel & Levine (2017)* proposed a model-agnostic meta-learning algorithm (MAML) (*Finn, Abbeel & Levine, 2017*; *Zheng et al., 2017*). Compared with the previous meta-learning method, this method introduces no additional parameters and has no restrictions on the model structure. Instead of updating a function or learning rule, it only uses gradient to update the learner weight. Andrei A. Liu et al. used the entire training set to pre-train the feature extractor in the case that MAML could not handle high-dimensional data well, and then used the parameter generation model to capture various parameters useful for task distribution (*Liu et al., 2020*; *Zheng et al., 2016a*). In contrast to the mamL-based improvement, applied antagonistic neural networks in the *Zhang et al. (2018)* field of meta-learning. This method uses the idea of generating antagonistic network to expand data and improve the learning ability of the network. However, the overall effect of this

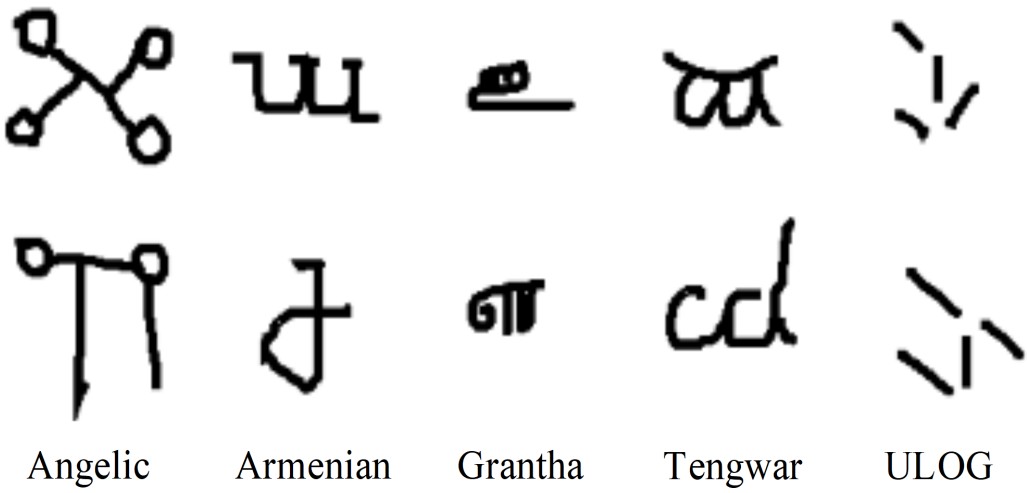

| Angelic | Armenian | Grantha | Tengwar | ULOG |

**Figure 1** Omniglot dataset example.

method is not stable, the relative cost is high, and the technology is not mature enough (*Zhang et al., 2018*; *Zheng et al., 2016b*).

The ability to learn and adapt quickly from small amounts of data is critical to AI. However, the success of deep learning depends to a large extent on a large amount of tag data, and in deep neural network learning, each task is isolated from the Learning, and Learning is always from scratch when facing a new task. Limited data or rapid generalization in a dynamic environment challenge current deep learning methods. Therefore, this paper takes the problem of image classification with few samples as the research object, combines measurement learning and optimization of initial representation methods, and designs a multi-scale meta-relational network. Finally, the classification accuracy and training speed of the two baseline data sets with small sample learning are improved.

## MATERIALS & METHODS

### Materials

#### Omniglot data set

In 2011, the Omniglot data set was collected by Brenden Lake and his collaborators at MIT through Amazon's Mechanical Turk (*Lake et al., 2011*; *Zheng et al., 2015*). It consists of 50 international languages, including mature international languages such as Latin and Korean, little-known local dialects, and fictional character sets such as Aurek-Besh and Klingon. The number of letters in each language varies widely, from about 15 to 40 letters, with 20 samples for each letter. Therefore, the Omniglot data set consists of 1623 categories and 32,460 images. Figure 1 illustrates the five languages of the Omniglot data set.

#### MiniImageNet data set

MiniImageNet data set, proposed by Vinyals et al., consists of 60,000 $84 \times 84 \times 3$ color images, a total of 100 categories, each with 600 samples (*Vinyals et al., 2016*). The distribution of data sets is very different, and the image category involves animals,

household goods, remote sensing images, food, etc. Vinyals did not publish this data set. Ravi and Larochelle randomly selected 100 classes from the ImageNet data set to create a new MiniImageNet data set, which was divided into training set, validation set and test set at a ratio of 64:16:20 (*Ravi & Larochelle, 2016*).

## Method
### *Metalearning based on metric Learning*

In meta-learning, metric learning refers to learning similarity measurement from a wide range of task Spaces, so that the experience extracted from the previous learning tasks can be used to guide the learning of new tasks to achieve the purpose of learning how to learn. The learner (meta-learner) learns the target set on each task of the training set by measuring the distance of the support set, and finally learns a metric. Then for the new task of the test set, it can quickly classify the target set correctly with the help of a small number of samples of the support set.

At present, the methods of small sample image classification based on metric Learning include: twin network, matching network, prototype network and relational network. The twin network is composed of two identical convolutional neural networks, and the similarity between two images is calculated by comparing the loss function with the input of paired samples. The other three methods, instead of using paired sample inputs, calculate the similarity between the two images by setting the support set and the target set. In this paper, the improvement is made on the basis of multi-scale relational network, so the main structure of this method is mainly introduced below.

The structure of relational network (*Sung et al., 2018*) first obtains feature graphs of support set and target set samples through embedded modules, then splices feature graphs of support set and target set samples in depth direction, and finally obtains relationship score by learning splicer features through relationship module, so as to determine whether support set and target set samples belong to the same category.

The calculation formula of relational fractions is as follows:

$$r_{i,j} = g_\phi(C(f_\varphi(x_i), f_\varphi(x_j))), i = 1, 2, \ldots, C \tag{1}$$

Where, $x_i$ represents the supporting set sample, $x_j$ represents the target set sample, $f_\varphi(x)$ represents the embedded module, $f_\varphi(x_i)$ and $f_\varphi(x_j)$ represent the feature graph of the supporting set and target set sample, and $C(f_\varphi(x_i), f_\varphi(x_j))$ represents the feature graph splicing operator. $g_\phi$ stands for relationship module.

### *Model-independent meta-learning algorithm*

According to the idea of transfer learning, when adapting to a new task, MAML only needs to fine-tune the learner so that the parameter of the learner is adapted from $\theta$ to $\theta'_i$. However, different from migration learning, MAML only needs to perform one or more gradient descent iterative steps on a small amount of data in new task $T_i$ to enable the learner to converge to the optimal parameter.

Here, $\theta'_i$ only considers the MAML that converges to the optimal parameter after an iterative step of gradient descent as follows:

$$\theta'_i = \theta - \alpha \nabla_\theta L_{T_i} f(\theta) \tag{2}$$

Where, $f(\theta)$ stands for learner, $L_{T_i}$ stands for loss on specific task, $\nabla_\theta L_{T_i} f(\theta)$ stands for loss gradient, and $\alpha$ stands for learner gradient update pace, namely learning rate of learner.

On a small amount of data in new task $T_i$, the learner can be converged to the optimal parameter $\theta_i'$ after one step of gradient descent iteration, so MAML must find a set of initialization representations that can be effectively fine-tuned according to a small number of samples. In order to achieve this goal, based on the idea of meta-learning "using previous experience to quickly learn new tasks", learners need to learn the learner parameter $\theta$ on different tasks. MAML defines this process as meta-learning process.

In the meta-learning process, for different training tasks, the optimal parameter $\theta_i'$ suitable for the specific task was obtained through a step of gradient iteration, and then sampled again on each task for testing, requiring $L_{T_i} f(\theta_i')$ to reach the minimum value. Therefore, MAML adopted the sum of test errors on different tasks as the optimization objective of the meta-learning process, as shown below:

$$min_\theta \sum_{T_i \sim p(T)} L_{T_i} f\left(\theta_i'\right) = \sum_{T_i \sim p(T)} L_{T_i} f\left(\theta - \alpha \nabla_\theta L_{T_i} f(\theta)\right) \tag{3}$$

where, $p(T)$ represents the distribution of the task set, $L_{T_i}$ represents the loss on the specific task, $\nabla_\theta L_{T_i} f(\theta)$ represents the loss gradient, and $\alpha$ represents the learning rate of the learner.

It can be seen from Eq. (3)–(3) that the optimization goal of the meta-learning process is to adopt the updated model parameter $\theta_i'$ adapted to the specific task, while the meta-optimization process is ultimately executed on the learner parameters $\theta$. The stochastic gradient descent method is adopted, and the updated iteration formula of model parameter $\theta$ is shown as follows:

$$\theta \leftarrow \theta - \beta \nabla_\theta \sum_{T_i \sim p(T)} L_{T_i} f\left(\theta_i'\right) \tag{4}$$

where, $p(T)$ represents the distribution of the task set, $L_{T_i}$ represents the loss on the specific task, $\nabla_\theta L_{T_i} f(\theta)$ represents the gradient, and $\beta$ represents the gradient update pace of the meta-learner, also known as the learning rate of the meta-learner.

Substituting Eq. (3)–(1) into Eq. (3)–(4), we can get:

$$\theta \leftarrow \theta - \beta \nabla_\theta \sum_{T_i \sim p(T)} L_{T_i} f\left(\theta - \alpha \nabla_\theta L_{T_i} f(\theta)\right) \tag{5}$$

### Algorithm design of multi-scale meta-relational network

In the multi-scale meta-relational network, we hope to find a set of characterization $\theta$ that can make fine adjustments efficiently according to a small number of samples. Where, $\theta$ is composed of feature extractor parameter $\varphi$ and metric learner parameter $\phi$. During the training process, each task is composed of training set $D_{train}$ and test set $D_{test}$. Where, $D_{train}$ is used for the inner optimization iteration, $D_{test}$ serves as the target set of the outer optimization iteration, and the support set adopts the support set $D_S$ in $D_{train}$.

In the inner optimization iteration process, on the new task $T_i$ with a small amount of data $D_{train}$, the learner converges to the optimal parameter $\theta_i$ after an iterative step of

gradient descent, as shown below:

$$\theta_i = \theta - \alpha \nabla_\theta L_{T_i,D_{train}} f(\theta) \tag{6}$$

Where $\alpha$ is a vector of the same size as $L'_{T_i,D_{train}}(\theta)$, representing the learning rate of the learner. The direction of the vector represents the update direction, and the magnitude of the vector represents the learning step.

For different training tasks, the optimal parameter $\theta_i$ suitable for specific tasks was obtained through a step of gradient iteration, and then tested on $D_{test}$ of different tasks, requiring $L_{T_i,D_{test},D_S} f(\theta_i)$ to reach the minimum value. Therefore, the meta-optimization objectives of the meta-training process are as follows:

$$min_\theta \sum_{T_i \sim p(T)} L_{T_i,D_s,D_{test}} f(\theta_i) \tag{7}$$

Substituting Eq. (3)–(6) into Eq. (3)–(7), we can find:

$$min_\theta \sum_{T_i \sim p(T)} L_{T_i,D_S,D_{test}} f\left(\theta - \alpha \nabla_\theta L_{T_i,D_{train}} f(\theta)\right) \tag{8}$$

For parameter $\theta$, the gradient calculation formula of formula (3–8) is as follows:

$$g_\theta = \frac{\partial}{\partial \theta} L_{T_i,D_S,D_{test}} f\left(\theta - \alpha \nabla_\theta L_{T_i,D_{train}} f(\theta)\right) \tag{9}$$

Substituting Eq. (3)–(6) into Eq. (3)–(9) can be simplified as follows:

$$
\begin{aligned}
g_\theta &= \frac{\partial L_{T_i,D_S,D_{test}} f(\theta_i)}{\partial f(\theta_i)} \times \frac{\partial f(\theta_i)}{\partial \theta_i} \times \frac{\partial \theta_i}{\partial \theta} \\
&= \frac{\partial L_{T_i,D_S,D_{test}} f(\theta_i)}{\partial f(\theta_i)} \times \frac{\partial f(\theta_i)}{\partial \theta_i} \times (I - \alpha \frac{\partial \left(\nabla_\theta L_{T_i,D_{train}} f(\theta)\right)}{\partial \theta})
\end{aligned} \tag{10}
$$

where $I$ is the unit vector.

It can be seen from Eq. (3)–(10) that the update process of the multi-scale meta-relational network involves the gradient calculation process, that is, when the gradient operator in the meta-target is used to propagate the meta-gradient, additional reverse transfer is required to calculate the Hessian vector product (*Kemp & Maas, 2009*).

In the multi-scale relational network, except for the sigMIod nonlinear function at the last full connection layer of the metric learner, all other nonlinear functions are ReLU. The ReLU neural network is almost linear in part, which indicates that the second derivative is close to zero in most cases. Therefore, the multi-scale element relational network, like MAML, ignores the second derivative in the calculation of the back propagation of the element gradient (*Kemp & Maas, 2009*), as follows:

$$
\begin{aligned}
g_\theta &= \frac{\partial L_{T_i,D_S,D_{test}} f(\theta_i)}{\partial f(\theta_i)} \times \frac{\partial f(\theta_i)}{\partial \theta_i} \\
&= \frac{\partial L_{T_i,D_S,D_{test}} f(\theta_i)}{\partial \theta_i} \\
&= \nabla_{\theta_i} L_{T_i,D_S,D_{test}} f(\theta_i)
\end{aligned} \tag{11}
$$

Therefore, in the process of calculating the outer gradient, the multi-scale element relation network stops the back propagation after calculating the gradient at $\theta_i$ and calculates the second derivative at $\theta$.

For parameter $\alpha$, the gradient calculation formula of formula (3–8) is as follows:

$$
\begin{aligned}
g_\alpha &= \frac{\partial}{\partial \alpha} L_{T_i, D_S, D_{test}} f\left(\theta - \alpha \nabla_\theta L_{T_i, D_{train}} f(\theta)\right) \\
&= \frac{\partial L_{T_i, D_S, D_{test}} f(\theta_i)}{\partial \theta_i} \times \frac{\partial \theta_i}{\partial \alpha} \\
&= -\nabla_\theta L_{T_i, D_{train}} f(\theta) \times \nabla_{\theta_i} L_{T_i, D_S, D_{test}} f(\theta_i)
\end{aligned}
\tag{12}
$$

In the outer optimization iteration process, stochastic gradient descent method is adopted, and the updated iteration formula of model parameter $\theta$ is shown as follows:

$$
\theta \leftarrow \theta - \beta \sum_{T_i \sim p(T)} \nabla_{\theta_i} L_{T_i, D_S, D_{test}} f(\theta_i)
\tag{13}
$$

Similarly, the update iteration formula of model parameter $\alpha$ is shown as follows:

$$
\alpha \leftarrow \alpha + \beta \sum_{T_i \sim p(T)} \left(\nabla_{\theta_i} L_{T_i, D_S, D_{test}} f(\theta_i)\right)\left(\nabla_\theta L_{T_i, D_{train}} f(\theta)\right)
\tag{14}
$$

In each inner iteration, all tasks can be used to update the gradient, but when the number of tasks is very large, all tasks need to be calculated in each iteration step, so the training process will be slow and the memory capacity of the computer will be high. If each inner iteration uses one task to update the parameters, the training speed will be accelerated, but the experimental performance will be reduced, and it is not easy to implement in parallel. So, during each inner iteration, we take num_inner_task for the number of tasks. By selecting the number of tasks in the inner iteration process as a super parameter, memory can be reasonably utilized to improve the training speed and accuracy.

During the meta-training of multi-scale meta-relational network, after the end of an epoch, the accuracy on the meta-validation set was calculated, and the highest accuracy on the meta-validation set was recorded so far. If consecutively N epochs (or more, adjusted according to the specific experimental conditions) did not reach the optimal value, it could be considered that the accuracy was no longer improved, and the iteration could be stopped when the accuracy was no longer improved or declined gradually. Output the model with the highest accuracy, and then test it with this model.

## EXPERIMENT AND RESULTS

### Multi-scale meta-relational network design

In the algorithm design of multi-scale meta-relational network:

1. Adopting the meta-SGD idea, the learning rate of the inner layer, that is, the learning rate of the learner, is taken as the learning vector and model parameter to learn together, so as to further improve the performance of the learner. 2. Considering that the small sample image classification method based on metric learning can be adapted to the new task without fine-tuning, the multi-scale meta-relational network adopts the MAML algorithm to learn and find the optimal parameters of the model in the meta-training process, and eliminates the inner gradient iteration in the meta-validation and meta-testing process.

**Table 1  Omniglot dataset experiment super parameter value.**

| Meta-learning rate $\beta$ | 1-shot | | 5-shot | | num_inner_task | |
|---|---|---|---|---|---|---|
| | $b_1$ | $b_2$ | $b_1$ | $b_2$ | 5-way | 20-way |
| 0.003 | 8 | 8 | 5 | 5 | 32 | 16 |

## Small sample learning benchmark dataset experiment

The model structure of multi-scale meta-relational network adopts multi-scale relational network, which is mainly composed of feature extractor and measure learner (*Biggs, 1985*). The second derivative is ignored in the back propagation process of the multi-scale element relational network. In practical implementation, the back propagation process can be truncated after the calculation of the outer gradient of the multi-scale element relational network is completed.

For experimental comparison, MAML and MetaSGD experiments are carried out simultaneously in this paper. According to *Finn, Abbeel & Levine (2017)* in the experiment of MAML and MetaSGD, a convolutional neural network composed of four layers of convolution and one layer of full connection is adopted as a learning device. Each convolution module of the learner is composed of a convolutional layer consisting of 64 filters of size 1, a batch standardization layer, a modified linear element layer and a maximum pooling layer of size 2. All the convolutional layers are filled with zero. The fourth convolution module is followed by the full connection layer. The number of neurons in the full connection layer is 64, and the nonlinear function log_SOFTmax is adopted. Different from the multi-scale relational network, the cross-entropy loss function is used to optimize the learning object.

### Omniglot data set

In the Omniglot data set experiment, all samples were processed into a size of 28×28 and a random rotation was used to enhance the data. From 1,623 classes, 1200, 211 and 212 classes were selected as meta-training set, meta-validation set and meta-test set respectively. In this section, single-sample image classification experiments of 5-ways 1-shot and 20-ways 1-shot and small-sample image classification experiments of 5-ways 5-shot and 20-ways 5-shot are conducted.

In the experiment, 100 episodes of multi-scale meta-relational network were taken as one epoch. The iteration times of the meta-training set, meta-validation set and meta-test set are accordingly 70,000, 500 and 500. Other super-parameter selections on the Omniglot dataset are shown in Table 1.

### Single sample image classification

It can be seen from Fig. 2 that, when the number of iterations is 42400, the multi-scale meta-relational network achieves the highest accuracy rate of 99.8667% in the 5-way 1-shot experiment on the meta-validation set. The iteration time of the multi-scale meta-relational network is less than that of the multi-scale relational network (method 1 in this paper). Therefore, compared with the convergence of the multi-scale relational network when the

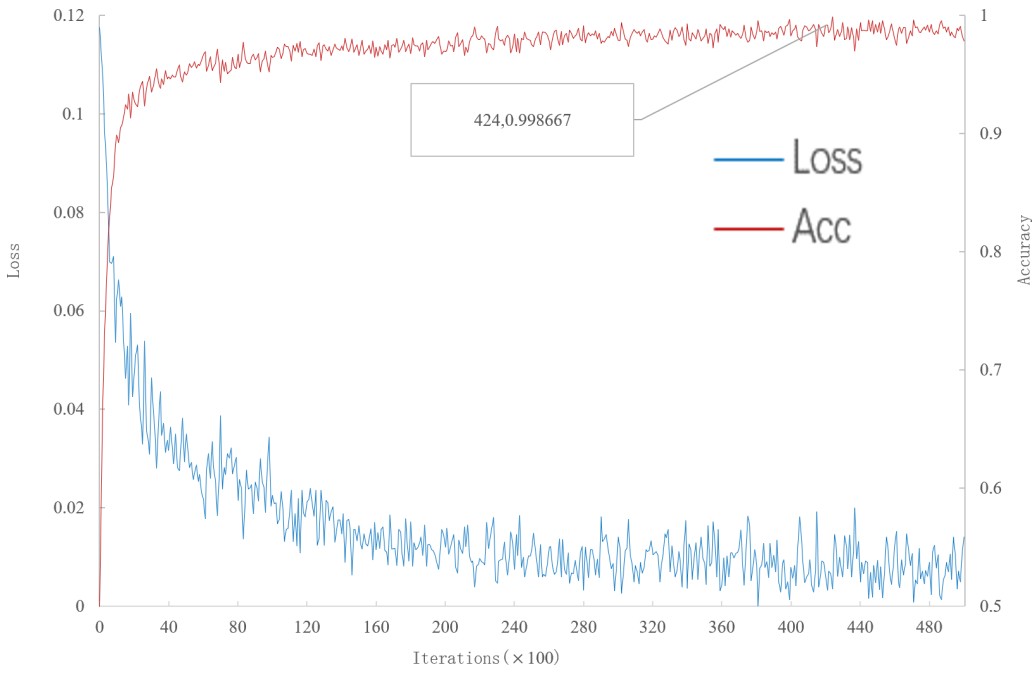

**Figure 2  Accuracy and loss iteration curves of 5-way 1-shot in multi-scale meta-relational network.**

**Table 2  Omniglot dataset single sample classification experimental results.**

| Model | Fine-tuning | Accuracy | |
|---|---|---|---|
| | | 5-way 1-shot | 20-way 1-shot |
| MAML | Y | 97.80 ± 0.32% | 95.60 ± 0.21% |
| Meta-SGD | Y | 99.50 ± 0.23% | 95.83 ± 0.36% |
| Multi-scale relational networks | N | 99.35 ± 0.25% | 97.41 ± 0.28% |
| Multi-scale meta-relational network | N | 99.57 ± 0.16% | 97.88 ± 0.20% |

iteration reaches 114,000, the learning speed of the multi-scale meta-relational network is faster than that of the multi-scale relational network.

The trained model was tested on the meta-test set, and the test results were shown in Table 2. According to Table 2:

1. The accuracy rate of the 5-way 1-shot experiment on the meta-test set of the multi-scale meta-relational network is higher than that of MAML and Meta-SGD, and about 0.22% higher than that of the multi-scale meta-relational network.

2. MAML and Meta-SGD based on optimized initial characterization need fine-tuning on new tasks, while the multi-scale relational network based on metric Learning can achieve good generalization performance on new tasks without fine-tuning. By comparing the two methods, the accuracy rate of the 5-way 1-shot experiment on the meta-test set of the multi-scale relational network is higher than MAML, but slightly lower than meta-SGD.

As can be seen from Fig. 3, when the number of iterations is 57,700, the multi-scale meta-relational network (method 2 in this paper) achieves the highest accuracy rate of

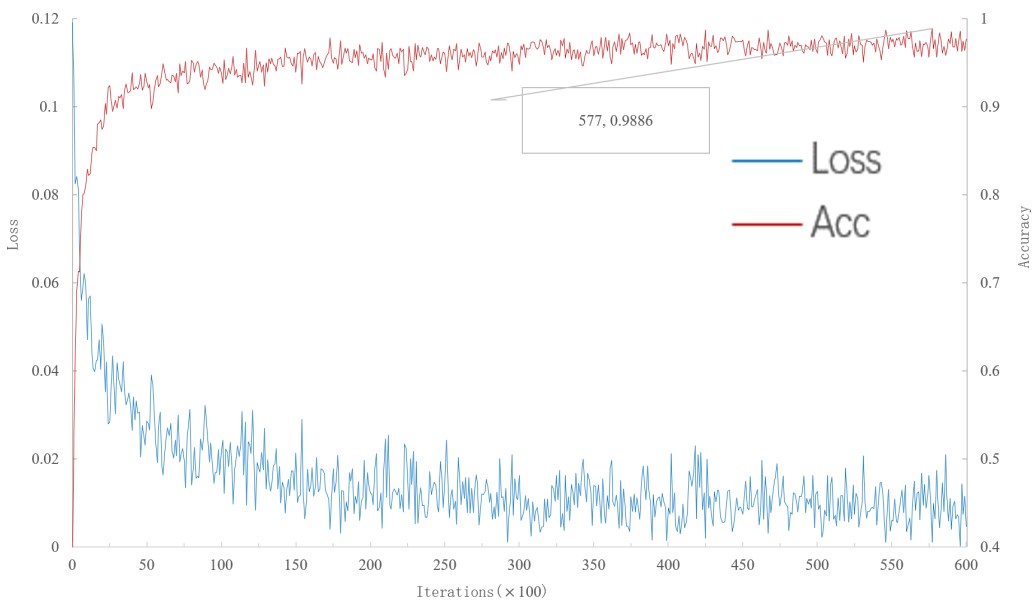

**Figure 3  Accuracy and loss iteration curves of 20-way 1-shot in multi-scale meta-relational network.**

98.86% in the 20-way 1-shot experiment on the meta-validation set. Compared with the multi-scale relational network (method 1 in this paper), the learning speed of multi-scale meta-relational network is faster than that of multi-scale relational network when it iterates to 89,000.

The trained model was tested on the meta-test set, and the results were shown in Table 2 below. According to Table 2:

1. The accuracy rate of the 20-way 1-shot experiment of the multi-scale meta-relational network on the meta-test set is about 0.47% higher than that of the multi-scale meta-relational network, which is higher than MAML and Meta-SGD.

2. MAML and Meta-SGD based on optimal initialization representation were compared with the multi-scale relational network based on metric Learning. The accuracy of the multi-scale relational network in the 20-way 1-shot experiment on the meta-test set was higher than that of MAML and meta-SGD.

### Small sample image classification

As can be seen from Fig. 4, when the number of iterations is 29100, the multi-scale meta-relational network (method 2 in this paper) achieves the highest accuracy of 99.89% in the 5-way 5-shot experiment on the meta-validation set. Compared with the multi-scale relational network (method 1 in this paper), the learning speed of the multi-scale meta-relational network is faster than that of the multi-scale relational network when it iterates to 293,500.

The trained model is tested on the meta-test set, and the results are shown in Table 3. According to Table 3:

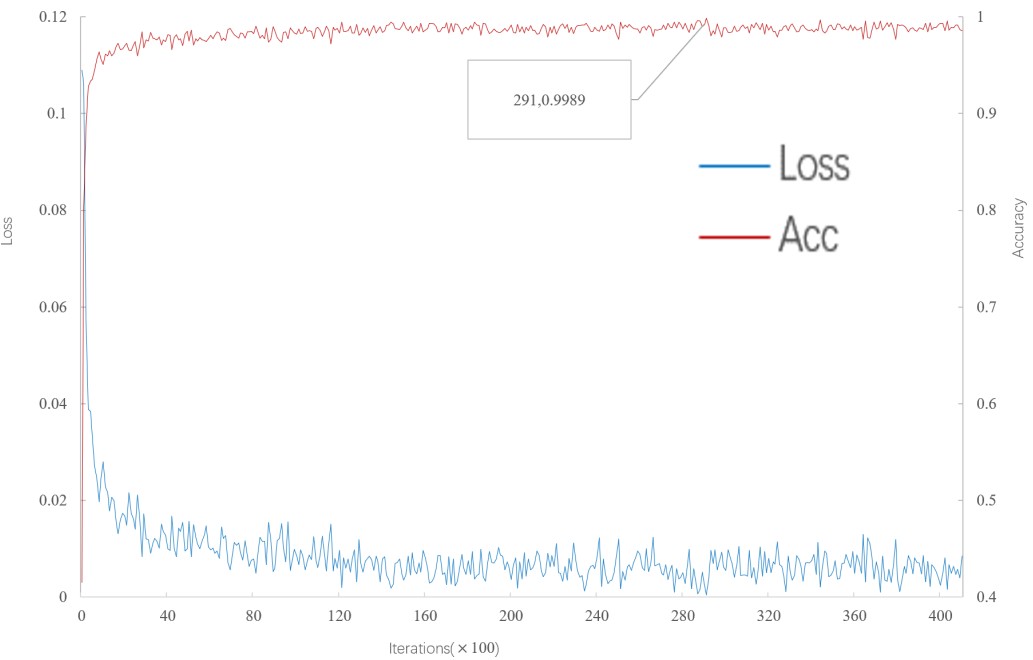

**Figure 4  Accuracy and loss iteration curves of 5-way 5-shot in multi-scale meta-relational network.**

**Table 3  Small sample classification experimental results in Omniglot dataset.**

| Model | Fine-tuning | Accuracy | |
|---|---|---|---|
| | | 5-way 5-shot | 20-way 5-shot |
| MAML | Y | 99.89 ± 0.10% | 98.90 ± 0.13% |
| Meta-SGD | Y | 99.91 ± 0.11% | 98.92 ± 0.22% |
| Multi-scale relational networks | N | 99.70 ± 0.08% | 99.01 ± 0.13% |
| Multi-scale meta-relational network | N | 99.84 ± 0.16% | 99.20 ± 0.12% |

1. The accuracy rate of the 5-way 5-shot experiment on the meta-test set of the multi-scale meta-relational network is about 0.14% higher than that of the multi-scale meta-relational network, but lower than MAML and Meta-SGD.

2. MAML and Meta-SGD based on optimal initial representation were compared with the multi-scale relational network based on metric learning. The accuracy of the 5-way 5-shot experiment on the meta-test set of the multi-scale relational network was lower than that of MAML and meta-SGD.

As can be seen from Fig. 5, when the number of iterations is 65800, the multi-scale meta-relational network (method 2 in this paper) achieves the highest accuracy rate of 99.82% in the 20-way 5-shot experiment on the meta-validation set. Compared with the multi-scale relational network (method 1 in this paper), the learning speed of multi-scale

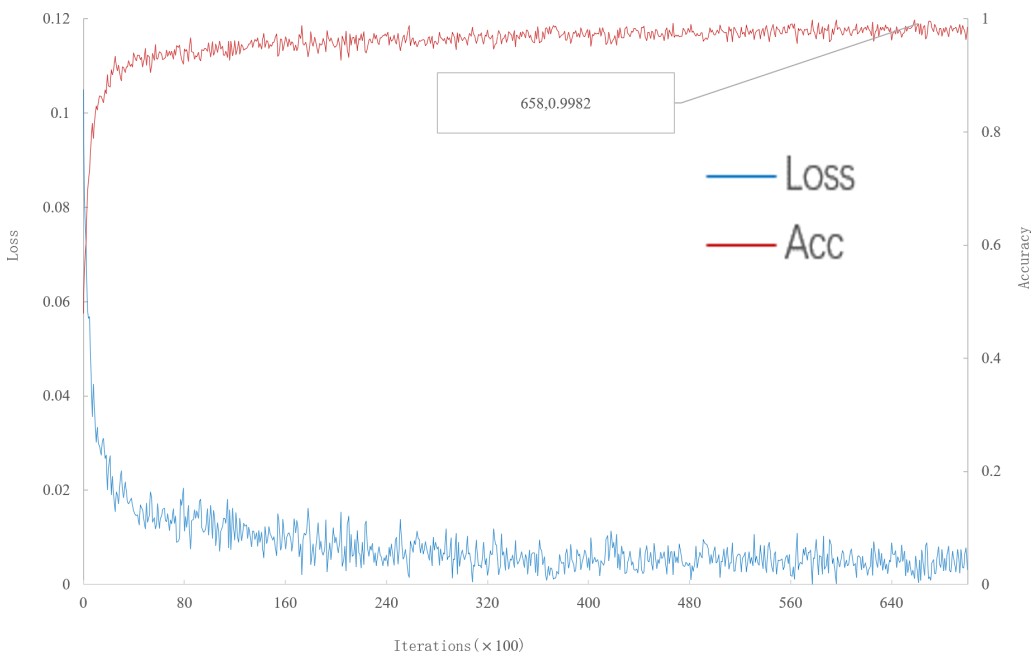

**Figure 5  Accuracy and loss iteration curves of 20-way 5-shot in multi-scale meta-relational network.**

meta-relational network is faster than that of multi-scale relational network when it iterates to 119500.

The trained model is tested on the meta-test set, and the results are shown in Table 3. According to Table 3:

1. The accuracy rate of the 20-way 5-shot experiment of the multi-scale meta-relational network on the meta-test set is about 0.19% higher than that of the multi-scale meta-relational network, which is also higher than MAML and Meta-SGD.

2. MAML and Meta-SGD based on optimized initial characterization were compared with the multi-scale relational network based on metric learning. The accuracy of the multi-scale relational network in the 20-way 5-shot experiment on the meta-test set was higher than that of MAML and meta-SGD.

In summary, the experimental results on Omniglot data set are as follows:

1. The classification accuracy of the 5-way 5-shot experiment on Omniglot data set was slightly lower than that of meta-SGD and MAML, and the other experiments were all higher than that of the three methods of multi-scale relational network, MAML and Meta-SGD. And the training speed of multi-scale meta-relational network is faster than that of multi-scale relational network.

2. MAML and Meta-SGD based on optimal initial characterization were compared with the multi-scale relational network based on metric learning. Except that the classification accuracy of 5-way 1-shot experiment on Omniglot data set was slightly lower than that of META-SGD and MAML, the classification accuracy of 5-way 5-shot experiment was lower than that of META-SGD and MAML, the other two groups of experiments were both higher than that of META-SGD and MAML.

**Table 4  miniImageNet dataset experiment super parameter value.**

| Meta-learning rate $\beta$ | 1-shot | | 5-shot | | num_inner_task | |
|---|---|---|---|---|---|---|
| | $b_1$ | $b_2$ | $b_1$ | $b_2$ | 1-shot | 5-shot |
| 0.003 | 10 | 10 | 5 | 5 | 8 | 4 |

**Table 5  The experimental results of 5-way 1-shot in MiniImagenet dataset.**

| Model | Fine-tunning | Accuracy |
|---|---|---|
| MAML | Y | $48.69 \pm 1.47\%$ |
| Meta-SGD | Y | $50.46 \pm 1.23\%$ |
| Multi-scale relational networks | N | $50.21 \pm 1.08\%$ |
| Multi-scale meta-relational network | N | $50.56 \pm 0.93\%$ |

## Miniimagenet data set

According to the ratio of 64:16:20, it is divided into meta-training set, meta-validation set and meta-test set. In this chapter, the classification experiment of single-sample image of 5-ways 1-shot and the classification experiment of small-sample image of 5-ways 5-shot is carried out.

In the experiment, the multi-scale meta-relational network took 500 episodes as one epoch. The number of iterations of the meta-training set, meta-validation set and meta-test set can be valued at 120,000, 600 and 600. Other super-parameter selections on the MiniImageNet dataset are shown in Table 4.

### *Single sample image classification*

As can be seen from Fig. 6, with the increase of iteration, loss decreases to convergence, and accuracy gradually increases to convergence. When the number of iterations is 87,000, the multi-scale meta-relational network (method 2 in this paper) achieves the highest accuracy rate of 51.234% in the 5-way 1-shot experiment on the meta-validation set. The iteration time of multi-scale meta-relational network is less than that of multi-scale relational network (method 1 in this paper). Therefore, compared with the convergence of multi-scale relational network when iteration reaches 155,000, the learning speed of multi-scale meta-relational network is faster than that of multi-scale relational network. The trained model is tested on the meta-test set, and the results are shown in Table 5. According to Table 5:

1. The accuracy rate of the 5-way 1-shot experiment on the meta-test set of the multi-scale meta-relational network is about 0.35% higher than that of the multi-scale meta-relational network, and higher than MAML and Meta-SGD.

2. MAML and Meta-SGD based on optimized initial characterization were compared with the multi-scale relational network based on metric learning. The accuracy of the 5-way 1-shot experiment on the meta-test set of the multi-scale relational network was higher than MAML, but lower than meta-SGD.

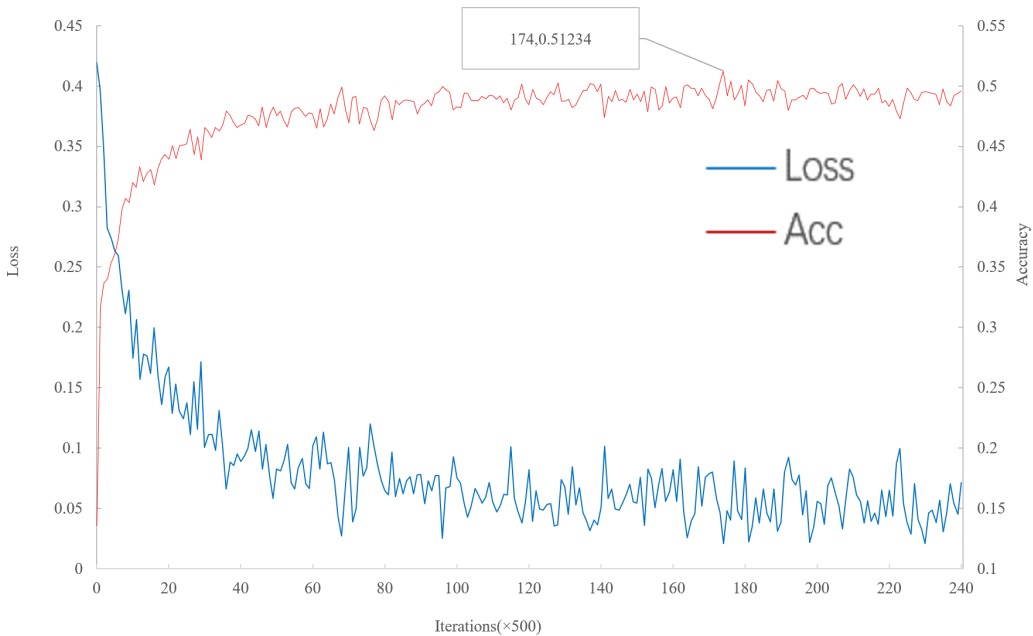

**Figure 6** Accuracy and loss iteration curves of 5-way 1-shot in multi-scale meta-relational network.

**Table 6** The experimental results of 5-way 5-shot in Miniimagenet dataset.

| Model | Fine-tunning | Accuracy |
| --- | --- | --- |
| MAML | Y | $63.10 \pm 0.82\%$ |
| Meta-SGD | Y | $64.03 \pm 0.66\%$ |
| Multi-scale relational networks | N | $65.89 \pm 0.32\%$ |
| Multi-scale meta-relational network | N | $66.23 \pm 0.55\%$ |

### *Small sample image classification*

As can be seen from Fig. 7, with the increase of iteration, loss decreases to convergence, and accuracy gradually increases to convergence. When the number of iterations is 62000, the multi-scale meta-relational network (method 2 in this paper) achieves the highest accuracy rate of 66.81% in the 5-way 5-shot experiment on the meta-validation set. Compared with the convergence of multi-scale relational network (method 1 in this paper) when iterative to 140,000, the learning speed of multi-scale meta-relational network is faster than that of multi-scale relational network.

The trained model is tested on the meta-test set, and the results are shown in Table 6. According to Table 6:

1. The accuracy rate of the 5-way 5-shot experiment on the meta-test set of the multi-scale meta-relational network is about 0.34% higher than that of the multi-scale meta-relational network, which is higher than MAML and Meta-SGD.

2. MAML and Meta-SGD based on optimized initial characterization were compared with the multi-scale relational network based on metric learning. The accuracy of the

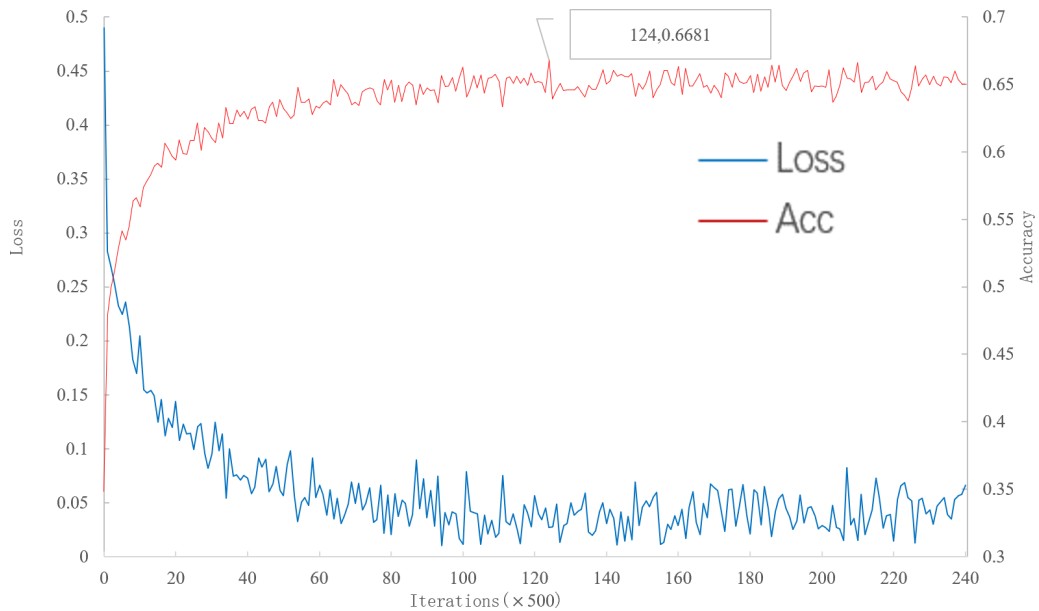

**Figure 7** Accuracy and loss iteration curves of 5-way 5-shot in multi-scale meta-relational network.

multi-scale relational network in the 5-way 5-shot experiment on the meta-test set was higher than that of MAML and meta-SGD.

In summary, the experimental results on Miniimagenet data set are as follows:

1. The classification accuracy of MNS on Miniimagenet data set is higher than that of MNS, MAML and META-SGD, and the training speed is higher than that of MNS.

2. By comparing the MAML and Meta-SGD based on optimal initial representation and the multi-scale relational network based on metric learning, the classification accuracy of the 5-way 1-shot experiment on Miniimagenet data set was slightly lower than that of meta-SGD, and the classification accuracy of the 5-way 5-shot experiment was higher than that of the META-SGD and MAML.

## DISCUSSION

The multi-scale meta-relational network proposed in this paper learns specific classification tasks by training the multi-scale relational network on a wide range of task Spaces. Then, MAML algorithm is used to find a set of highly adaptive parameters, which can make good use of the experiential knowledge learned in previous tasks and realize the ability of Learning.

1. By comparing the MAML and Meta-SGD based on optimized initial characterization with the multi-scale relational network based on metric learning, the overall experimental performance of the two methods is almost the same in terms of accuracy, and it cannot be absolutely determined that one method is better than the other.

2. MAML, Meta-SGD and multi-scale meta-relational networks based on optimal initial characterization are compared separately, although Meta - SGD and MAML on 5 - way

Omniglot data set the classification accuracy of cyber-shot experiment was slightly higher than that of multi-scale mata relation network, but Meta - SGD and MAML need to use a small amount of data from a new task computing steps to update one or more of the gradient parameters, on a new task has the largest generalization performance, and multi-scale meta network for not seen in the process of training the new category of image, With the help of a small number of samples for each new category, it has a good generalization ability. Therefore, the method combining metric learning and learning optimization initial representation has higher performance than the method based on optimization initial representation.

3. Compare the multi-scale relational network based on metric learning with the multi-scale meta-relational network separately. Multi-scale meta network will study MAML initialization characterization method is introduced to measure the Learning, because the initialization characterization is more suitable for MAML learning Yu Zaiyuan task distribution of training data sets, thus reduced multi-scale mate network data sets the influence of the difference of distribution in the small sample experiment results on benchmark data set higher than that of multi-scale network, network learning speed and multi-scale relation in multi-scale network. Therefore, the method combining metric learning and learning optimization initial representation has higher performance than the method based on metric learning.

Therefore, the method combining metric learning and optimal learning initial representation has higher performance than the method based on metric or optimal initial representation. The multi-scale meta-relational network enables the learned measurement method to have stronger generalization ability, which not only improves the classification accuracy and training speed on the benchmark set, but also avoids the situation that MAML needs fine-tuning.

## CONCLUSION

Considering the difference of task set distribution, and in order to make the learned measurement have stronger generalization ability, this paper designs a multi-scale meta-relational network (MSNN), a classification method based on optimized initialization representation, for image with few samples. First, the multi-scale meta-relational network adopts the meta-SGD idea and takes the inner learning rate as the learning vector and model parameter to learn together. Secondly, in the process of meta-training, the multi-scale meta-relational network adopts the MAML algorithm to learn and find the optimal parameters of the model, while in the process of meta-validation and meta-test, the inner gradient iteration is eliminated and the test is carried out directly. The experimental results show that the method combined with metric learning and optimized initial representation has higher performance than the method based on metric or optimized initial representation. The multi-scale meta-relational network enables the learned measurement method to have stronger generalization ability, which not only improves the classification accuracy and training speed on the benchmark set, but also avoids the situation that MAML needs fine-tuning.

Although the method in this paper improves the classification accuracy on the learning benchmark set with few samples and the overfitting situation, it still needs to be improved in the following aspects:

1) Compared with miniImageNet datasets, the Omniglot dataset is simpler; on the small sample learning problems, Omniglot baseline is above 97%. However, the classification of the dataset on miniImageNet dataset classification effect is not very ideal. Finding a better way of meta learning in order to reduce the influence of the difference of the task set ascend in the miniImageNet data set classification effect is worth exploring.

2) In the multi-scale element relational network, partial gradient information will be lost when the second derivative is omitted. Whether a better way to simplify the second derivative can be found is also the direction to be solved in future work.

## ACKNOWLEDGEMENTS

The authors express their sincere appreciation and profound gratitude to research assistant Xia Tian, WeiZheng Huang and LiHong Song, for their helping and supporting on collection and sorting of the data.

### Funding

This work was supported by the Sichuan Science and Technology Program (2019YJ0189). The funders had no role in study design, data collection and analysis, decision to publish, or preparation of the manuscript.

### Grant Disclosures

The following grant information was disclosed by the authors:
Sichuan Science and Technology Program: 2019YJ0189.

### Competing Interests

The authors declare there are no competing interests.

### Author Contributions

- Wenfeng Zheng analyzed the data, prepared figures and/or tables, authored or reviewed drafts of the paper, and approved the final draft.
- Xiangjun Liu performed the experiments, performed the computation work, prepared figures and/or tables, and approved the final draft.
- Lirong Yin conceived and designed the experiments, analyzed the data, authored or reviewed drafts of the paper, and approved the final draft.

### Data Availability

The code for the relation network used in the article is available in the Supplemental File.

## Supplemental Information

Supplemental information for this article can be found online at http://dx.doi.org/10.7717/peerj-cs.613#supplemental-information.

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
