# Peer review of "Research on image classification method based on improved multi-scale relational network"

_PeerJ Computer Science, doi:10.7717/peerj-cs.613_

## Round 0.1 · original submission · Major Revisions

The authors should address all review comments to revise the manuscript. The revised manuscript could be resubmitted for further review.

Reviewer 1 ·

Basic reporting

This paper proposes a new approach of multi-scale meta-relational network as a classification approach based on optimized initialization representation for image with few samples. The idea of META-SGD is adopted and model-independent meta-learning algorithm is introduced to find the optimal parameters of the model. The contents and target of the paper take interests of readers.

Experimental design

Experiments suggests the performance according to the theoretical aspect discussed in the paper. Experimental parts satisfies the necessary comparison between models and tasks.

Validity of the findings

The proposed approach of meta-learning is very important topic in the recent deep learning research since the principle is based on the idea “how to learn by using previous experience” with the small number of sample images.

Additional comments

Meta-learning of multi-scale meta-relational network proposed in this paper has advantage to have the highest accuracy as the meta-validation set is recorded. Paper suggests that MAML and Meta-SGD need fine-tuning on new tasks while multi-scale relational network based on metric learning can achieve good generalization performance on new tasks without fine-tuning.
Experiments suggests the performance according to the theoretical aspect discussed in the paper. Detailed descriptions are provided and the paper is well written in both of theoretical aspect and experimental part.

Reviewer 2 ·

Basic reporting

The title does not reflect the contribution of this paper.
New learning and classification methods are introduced and not research on mage classification method
The paper proposes a new classification approach addressing small dataset problems. The derivations of the proposed method is provided. The approach is evaluated with standard datasets and measures.

Suggestions for improvements on the paper for the presentation issues are given below.

1. Chronology is important. However, it is redundant to keep mentioning the years as citations include publication year.-- Example. In 2016, .... (citation, 2016)

2. Quite often though capital letters and small letters are used inconsistently.
ARTIFICIAL intelligence
deep Learning
This method USES the idea of…..

3. Consistencies in citations convention is needed in the presentation

4. The last reference on the reference list is incomplete ….

Experimental design

Yes, a clear comparative evaluation was provided

Validity of the findings

All datasets and measures were clearly provided.

Additional comments

All my comments largely on the presentation was given above.

The following sections of the paper needs further clarification

1. Sentence 219 n the multi-scale meta-relational network, we hope to find a set of characterization (thetha) that can make fine adjustments efficiently according to a small number of samples. Where, is composed of feature extractor parameter and metric learner parameter . These two parameters are mentioned but not explained. Elaboration on these are needed

2. Sentence 517 at the end --- a better way of yuan learning
- Is this referring to the ----MAML learning Yu Zaiyuan task distribution of …
- Elaboration on yuan learning is needed

Reviewer 3 ·

Basic reporting

the processing of iterative is clear, but the convergence and convergence speed are not involved.

Experimental design

no comment

Validity of the findings

no comment

Additional comments

to classify image based on small sample, a novel parameters iteration method to improve multi-scale meta-relational network is presented in this paper. the classification results illustrate the superiority of the method. but there are still some problem existed.
1. there are some mistakes in serial number of formula. For example: line 237: Equation (3-6) and (3-7) should be (6) and (7);
2. the processing of iterative is clear, but the convergence and convergence speed are not involved.
3. the results is better than existing methods, are the differences significant enough? will the gap depend on the sample?

---

## Round 0.2 · accepted · Accept

The reviewer recommended accepting the manuscript.

Reviewer 3 ·

Basic reporting

no comment

Experimental design

no comment

Validity of the findings

no comment

Additional comments

Authors have made suitable modifications to the suggestions